# The Impact of Biliary Injury on the Recurrence of Biliary Cancer and Benign Disease after Liver Transplantation: Risk Factors and Mechanisms

**DOI:** 10.3390/cancers16162789

**Published:** 2024-08-07

**Authors:** Chase J. Wehrle, Rebecca Panconesi, Sangeeta Satish, Marianna Maspero, Chunbao Jiao, Keyue Sun, Omer Karakaya, Erlind Allkushi, Jamak Modaresi Esfeh, Maureen Whitsett Linganna, Wen Wee Ma, Masato Fujiki, Koji Hashimoto, Charles Miller, David C. H. Kwon, Federico Aucejo, Andrea Schlegel

**Affiliations:** 1Transplantation Center, Cleveland Clinic, Cleveland, OH 44195, USA; wehrlec@ccf.org (C.J.W.); aucejof@ccf.org (F.A.); 2Department of Inflammation and Immunity, Lerner Research Institute, Cleveland Clinic, Cleveland, OH 44195, USA; panconr@ccf.org (R.P.); jiaoc@ccf.org (C.J.);; 3General Surgery and Liver Transplantation Unit, IRCCS Istituto Tumori, 20133 Milan, Italy; 4Department of Gastroenterology and Transplant Hepatology, Cleveland Clinic, Cleveland, OH 44195, USA; 5Novel Therapeutics Center, Taussig Cancer Institute, Cleveland Clinic, Cleveland, OH 44195, USA

**Keywords:** liver transplantation, biliary complications, cholangiocarcinoma, benign disease recurrence

## Abstract

**Simple Summary:**

Ischemia-reperfusion injury (IRI) is the source of significant graft inflammation in liver transplantation. Such injuries can lead to increased recurrence of both benign and malignant diseases of the biliary tree after transplant. Herein, we review the mechanisms and risk factors for the link between IRI and biliary recurrence, as well as evidence for strategies to mitigate this damage.

**Abstract:**

Liver transplantation is known to generate significant inflammation in the entire organ based on the metabolic profile and the tissue’s ability to recover from the ischemia-reperfusion injury (IRI). This cascade contributes to post-transplant complications, affecting both the synthetic liver function (immediate) and the scar development in the biliary tree. The new occurrence of biliary strictures, and the recurrence of malignant and benign liver diseases, such as cholangiocarcinoma (CCA) and primary sclerosing cholangitis (PSC), are direct consequences linked to this inflammation. The accumulation of toxic metabolites, such as succinate, causes undirected electron flows, triggering the releases of reactive oxygen species (ROS) from a severely dysfunctional mitochondrial complex 1. This initiates the inflammatory IRI cascade, with subsequent ischemic biliary stricturing, and the upregulation of pro-tumorigenic signaling. Such inflammation is both local and systemic, promoting an immunocompromised status that can lead to the recurrence of underlying liver disease, both malignant and benign in nature. The traditional treatment for CCA was resection, when possible, followed by cytotoxic chemotherapy. Liver transplant oncology is increasingly recognized as a potentially curative approach for patients with intrahepatic (iCCA) and perihilar (pCCA) cholangiocarcinoma. The link between IRI and disease recurrence is increasingly recognized in transplant oncology for hepatocellular carcinoma. However, smaller numbers have prevented similar analyses for CCA. The mechanistic link may be even more critical in this disease, as IRI causes the most profound damage to the intrahepatic bile ducts. This article reviews the underlying mechanisms associated with biliary inflammation and biliary pathology after liver transplantation. One main focus is on the link between transplant-related IRI-associated inflammation and the recurrence of cholangiocarcinoma and benign liver diseases of the biliary tree. Risk factors and protective strategies are highlighted.

## 1. Introduction

Liver transplantation (LT) is a life-saving intervention for patients with end-stage liver disease [1,2]. Post-transplant outcomes generally continue to improve, yet a supply–demand mismatch between available grafts and the high number of candidates remains, together with a greater-than-ideal rate of graft-specific complications [3,4,5]. The optimal strategy to address the supply–demand gap is to increase the utilization of riskier grafts, yet this is limited primarily by the risk of graft-related complications, such as the development of biliary strictures. This condition is mediated by mitochondrial injury, the primary trigger of ischemia-reperfusion injury (IRI), leading to persistent tissue inflammation and an increased risk of biliary strictures, which is a significant cause of early graft loss within 6–12 months after LT.

Transplant oncology is an ever-emerging field addressing the increasing rates of primary and secondary hepatic malignancies [6,7,8]. Cholangiocarcinoma (CCA) is the second most common primary liver cancer and is highly lethal. Liver transplantation is an effective treatment for both intrahepatic (iCCA) and perihilar (pCCA) cholangiocarcinoma, though recurrence rates remain high despite improved patient selection.

While IRI inflammation is well-recognized as negatively impacting transplant outcomes, it has also been linked to an increased risk of recurrence of cancer and benign disease after transplant [9,10,11,12,13]. This link has primarily been described in LT for hepatocellular carcinoma (HCC), likely due to its higher prevalence. However, transplant-associated inflammatory damage is most prominent in the biliary system, and thus might have the greatest impact on LT for CCA.

This work reviews the mechanisms behind biliary damage in liver transplantation, and the link between transplant-associated inflammation and cancer recurrence. Additionally, we aim to provide a mechanistic link between biliary inflammation and cholangiocarcinoma, and to explore strategies surgeons might use to mitigate this biliary injury in modern practice [14].

## 2. Biliary Inflammation, Cholangiocarcinoma, and Liver Transplantation

This study is a narrative review. A search strategy was initiated by querying keywords including “liver transplant”, “cholangiocarcinoma”, “biliary disease”, “ischemia-reperfusion injury”, and “disease recurrence” in the database PubMed^®^. Articles were compiled and reviewed. Additionally, the Cited by and References sections of selected articles were reviewed, and relevant articles from these sections were also included.

### 2.1. A Mechanistic Overview of Ischemic Cholangiopathy

The deceased donor transplant process involves inherent injury processes that have been extensively researched [15]. At organ procurement, grafts are exposed to hypoxia under cold conditions (i.e., cold ischemia time (CIT)) at approximately 4 °C to minimize the cellular metabolic activity and related ischemic damage [15]. The graft remains cold until implantation, when it is exposed to recipient warm ischemia time (rWIT), becoming more metabolically active at higher temperatures when compared to the ice box. Donation after Circulatory Death (DCD) grafts experience an additional injury after treatment withdrawal and during procurement, prior to pronouncing the donor’s death. This period of hypoxia (i.e., functional donor warm ischemia time) follows various definitions and is calculated from low donor saturation (pO_2_ < 80%) or hypotension (systolic blood pressure < 80 mmHg or MAP < 50 mmHg) due to cold in situ donor flush [16,17,18].

Though toxic metabolites do accumulate during ischemia, the most critical damage arises when oxygen is reintroduced. This entire cycle, termed ischemia-reperfusion injury (IRI), can be linked to many of the biliary complications associated with liver transplantation, in addition to other systemic effects such as post-reperfusions syndrome [19,20]. Organ flush quality and transplant-induced biliary stasis play additional key roles, with the latter generating greater exposure to cytotoxic bile salts that can exacerbate biliary inflammation [21].

Ischemic cholangiopathy, frequently termed the “Achilles heel” of liver transplantation, has long hindered donor pool expansion and caused severe patient morbidity and mortality [22,23]. Ischemic cholangiopathy is defined as a biliary stricture above the anastomosis, known as a non-anastomotic stricture (NAS), without a coinciding major vascular complication [24]. NAS typically present first clinical signs within 2 weeks to 1 year post-transplant, with a median onset at 4.1 months [25]. Croome et al. described four NAS sub-types. Diffuse necrosis and multi-focal progressive forms are those generally associated with graft loss, with >70% requiring retransplantation within the first year [26]. The other two sub-types (i.e., confluence dominant and “minor”) are clinically less severe and may resolve with the need for endoscopic interventions and readmissions, linked to significant cumulative morbidity and increased costs [26,27].

For most of the history of liver transplantation, the etiology of NAS has been unknown. As a result, predicting its occurrence remains challenging, and DCD liver grafts are frequently discarded with the “quasi-scientific” belief that their use might predispose recipients to NAS development. The first and most consistently reported risk factors for NAS have been prolonged total, functional, and asystolic donor warm ischemia times (dWIT) [28,29]. Expectedly, cold ischemic time (CIT) is also frequently listed as contributor, though there have been many proposed thresholds for predicting NAS, none of which is definitive [30,31]. Finally, graft-specific factors (i.e., donor BMI and macrosteatosis) and recipient factors (i.e., prolonged ICU stay and vasopressor requirements) were described to affect NAS rates, lacking uniform thresholds for any of these factors (Figure 1) [32,33,34,35].

How do all these individual clinical risk factors combine to damage the biliary tree? And, perhaps as critically, why do these factors lead to NAS in some patients, yet not in others?

The answer to both questions lies in the metabolic condition and mitochondrial injury of the transplanted graft. More specifically, there may be two causes of such ductal strictures: either ongoing mitochondrial-mediated inflammation, which leads to fibrotic tissue deposition, or initial IRI-associated irreversible biliary stem cell damage, leading to scar formation [36]. The IRI cascade is a cycle of liver cell injury during transplantation that begins in the donor based on the cellular metabolism. The duration of donor warm and cold ischemia while the liver is transported is the next contributor. During dWIT, the liver is metabolically active, but lacks appropriate oxygen supply, which rapidly consumes cellular ATP and generates potentially harmful TCA cycle products, such as succinate [23,37,38]. The cooling of the liver minimizes the need for ATP use in the graft, slowing down the respiratory chain, yet the graft continues to accumulate injury in the absence of oxygen. Subsequent oxygen reintroduction (reperfusion) leads to the reactivation of the respiratory chain and TCA cycle, as mitochondria aim to reduce accumulated toxic metabolites. However, this occurs too rapidly. High succinate levels overwhelm the mitochondrial complex II, with a subsequent undirected and retrograde electron flow that triggers reactive oxygen species (ROS) production and release from complex I, instigating the entire IRI cascade [39]. It has long been established that bile ducts are the most susceptible to hypoxia and associated complications, preferentially predisposing the components of the biliary tree to ischemia-related complications [40,41].

These mechanisms are not unique to liver transplantation, and also mediate damage in myocardial infarction, stroke, and numerous other situations where tissue ischemia is induced [42]. In these conditions, strategies to minimize the accumulation of succinate under ischemic conditions have mitigated the production and damage of pro-inflammatory ROS associated with too rapid downstream succinate oxidation [37].

Let us consider, then, two hypothetical DCD grafts exposed to the same theoretical transplant injury of 40 min functional dWIT, 10 min aystolic warm ischemia time, and 6 h CIT. Why will one graft develop IC, while another liver will function for years without the need for any intervention at the biliary tree? In short, grafts have differential inherent metabolic resistance patterns to both IRI and the initial ROS generation. This metabolic tolerance stems from differences in the function of the mitochondrial respiratory chain and, predominantly, complex I. Of note, together with ROS, complex I releases Flavin Mononucleotide (FMN) immediately when oxygen is reintroduced in ischemic tissues. FMN is a marker of mitochondrial dysfunction and was found in all organs at reperfusion, during transplantation, after stroke or myocardial infarction, and during machine perfusion [38,43,44,45,46,47]. Both ROS and FMN molecules are rapidly released within the first few minutes. Of note, while ROS are difficult to quantify and disappear quickly, FMN can be easily measured using fluorescence spectroscopy [43,44]. Surrounding cells respond with damps and cytokine release to this IRI instigator, thereby maintaining an inflamed status that expands through initially unaffected cells to the entire organ and recipient system.

### 2.2. Inflammation Triggered by Ischemia-Reperfusion Injury and Cancer Recurrence

Interestingly, this pro-inflammatory mechanism does not just mediate biliary stricturing. This should be unsurprising, as the ROS and damage-associated molecular patterns (DAMPs) do not just reside in the cholangiocytes but are created throughout the graft [23]. This includes the aforementioned pathways, along with the activation of the NLR-pyrin-domain-3 (NLRP-3)-mediated “inflammasome”, which activates caspase-3-mediated apoptosis, and upregulates the secretion of IL-1β and IL-18, which activate the NF-kB pathway [48,49]. Additionally, inflammation as a response to ischemia leads to complement protein C3a and C5a activation, and platelet activation in hepatic sinusoids [50].

These inflammatory cascades lead to the described graft damage but, unfortunately, the potential detriments do not end there. Mitochondrial-initiated inflammation is linked to the development of a pro-tumorigenic milieu, which, in turn, supports classically defined cancer recurrence pathways [51]. This link occurs in multiple ways, including the promotion of circulating tumor cell (cTC) depositions, attachment to endothelial structures, the invasion and proliferation in the newly implanted liver graft, the direct contribution of ROS to mutagenesis in tumor cells, and the promotion of cellular metastasis via mitochondrial signaling [52,53].

These effects are not just local in the liver graft. IRI creates systemic inflammatory changes, including the activation of the innate immune system, which can manifest clinically as a greater propensity for acute cellular rejection [50,54]. Systemic and ongoing inflammation leads to significant phenotypic changes locally, such as in the tumor microenvironment (TME), and systemically, which, in turn, promotes the distant regrowing of the tumor (i.e., in the lung), and its local proliferation and migration, through the resettling of cTCs attracted by the inflammatory environment [55]. Finally, sinusoidal epithelial cells (SECs) regulate the cellular and fluid permeability of the hepatic sinusoids. Prolonged hypoxia induces hypoxia-inducible factor 1α (HIF-1α) upregulation, which, in turn, promotes angiogenesis, and tumor cell permeation and proliferation, mediating cancer recurrence [9,56]. Indeed, Van der Bilt et al. demonstrated in 2005 that IRI promotes the proliferation of hepatic micro-metastases using mouse models, which was then confirmed in various follow-up studies in rodents [9,10,11].

A direct clinical link between graft-mediated inflammation and cancer recurrence has been demonstrated very elegantly by both Parente et al. and Kim et al., who each demonstrated that the use of very small liver grafts (graft-to-recipient weight ratio, GRWR, <0.8 or <0.7) is associated with impaired oncologic and patient outcomes after living donor liver transplant (LDLT) for hepatocellular carcinoma (HCC) [13,57]. This occurs, as described in the Parente study, through a greater pro-inflammatory molecule accumulation in smaller livers, as these grafts require greater proliferation to sustain recipient metabolic needs [13,58].

Mitochondria are, again, the key link between the greater proliferation seen in small grafts and the associated higher oncologic risk. Specifically, after LDLT and liver resection, hepatic oval cells (stem cells) begin to divide as part of the liver’s inherent regenerative capacity [59,60,61]. Such division relies on mitochondrial ATP generation, as regeneration will be quite lacking. This begins with the simultaneous activation of the Wnt/β-catenin and Notch signaling pathways, as well as the stimulation of LPS-mediated Kupfer cell activation [62]. The latter leads to the further release of the pro-inflammatory molecules maintaining IRI, specifically activating TNFα and IL-6, which subsequently activate hepatocyte proliferation via cyclin pathways [62,63]. Hepatocyte proliferation, mediated by HIF, VEGF, and the JAK/STAT and Ras/MAPK pathways, is another direct consequence [62,64,65]. A smaller graft leads to a greater activation of such pro-inflammatory cascades, a higher IRI degree, and a more significantly pro-tumorigenic milieu. Unfortunately, drivers of hepatic proliferation also promote tumor proliferation, leading to a greater cancer recurrence risk with more injury. Specific to CCA, in addition to the aforementioned mechanisms, the Wnt/β-catenin pathway is integrally linked to the Hippo/Yap/Taz pathway, and the two are co-upregulated in cases of active hepatocyte regeneration [62,66]. Hippo/Yap/Taz has been linked to biliary neoplastic proliferation, and its dysregulation is specifically implicated in CCA [67].

These inflammatory cascades can be mediated by surgical factors, such as portal vein flow, graft size, and more. This concept is critical in LDLT, where both too much and too little portal vein flow can be detrimental. This is based on the presence of NO and other inflammatory mediators in the Disse space [68]. Portal vein flow also mediates shear stress-induced inflammation, alters endothelial cell signaling, and affects downstream inflammatory mediators [69]. Many such molecules are pro-proliferative, thus mediating graft regeneration, and smaller grafts will have greater portal flows and pressures, triggering higher pro-regenerative signals. Although this is a protective mechanism benefitting LD recipients, the elevated risk for cancer recurrence and biliary structuring appears to be of high relevance as well [62,70].

### 2.3. Biliary Inflammation and Biliary Cancer Recurrence

There is a clear link between IRI and cancer recurrence, which has been demonstrated mechanistically and clinically in HCC [71,72]. However, as discussed in Section 1 and Section 2, some of the most critical pro-inflammatory IRI effects appear in the biliary system. So, what about the recurrence of biliary cancers?

CCA can be divided into extrahepatic (eCCA), intrahepatic (iCCA), and peri-hilar (pCCA), the latter two of which can be treated with liver transplantation. pCCA is the most common type in Western countries, accounting for ~60% of cases, followed by eCCA (20%) and iCCA (20%) [73]. CCA overall has an incidence rate of 0.3–6 per 100,000 persons per year. However, CCA has a uniquely high incidence rate (>6 per 100,000 persons) in Southeast Asia [73].

Understanding the pathogenesis of cholangiocarcinoma remains challenging, with various oncogenic pathways involved (Figure 2). These include the Notch and Wnt/β-catenin pathways, both implicated in CCA development. Genetic instability also plays a role, with alterations in MYC oncogene pathways. Other metabolic changes, including lipid metabolism and fatty acid oxidation, are enhanced in CCA, while pro-inflammatory cytokines, like IL-6 and IL-8, contribute to tumorigenesis. Angiogenesis, mediated by VEGF and other molecules, is also a key contributor in CCA. Additionally, cancer stem cells (CSCs) are implicated in recurrence and drug resistance, with ties to epithelial–mesenchymal transition (EMT) [74,75].

In the context of CCA, IRI has been shown to significantly impact tumor progression. Necroptosis, a form of programmed cell death, is particularly implicated in promoting oncogenic processes in CCA. Necroptotic inflammation can lead to chromatin changes that are more prominent in CCA, facilitating tumor progression [76,77]. Further, ongoing cellular necrosis, which is a hallmark of biliary sloughing in NAS, leads to angiogenesis and the promotion of tumor-associated macrophages. This has, in turn, been linked to rapid iCCA progression [78]. Another critical factor in IRI-related CCA progression is the upregulation of hypoxia-inducible factor 1-alpha (HIF-1α). This molecule plays a significant role in adapting the tumor microenvironment to hypoxic conditions, enhancing tumor survival and promoting metastasis. The hypoxic conditions induced by IRI lead to HIF-1α activation, which, in turn, promotes glycolysis, angiogenesis, and other pathways that support tumor growth and spread [79]. Various studies have suggested that targeting these pathways could offer new therapeutic strategies. For instance, inhibiting HIF-1α or modulating necroptosis-related proteins might reduce the oncogenic potential of IRI and improve outcomes for patients with biliary cancers [80]. Such mechanisms have also been demonstrated in remote ischemic preconditioning (RIPC) and other in-flow pedicle clamping approaches, which have also been shown to upregulate HIF′ HIF-1α [81,82].

Overall, the interplay between IRI and the recurrence of biliary cancers like CCA involves complex mechanisms including necroptosis, chronic inflammation, and hypoxia-induced metabolic reprogramming, all of which contribute to the more aggressive nature of these tumors. Further research could dive into such mechanisms and the role of novel therapeutic interventions aimed at mitigating IRI-induced cancer recurrence.

## 3. Inflammation and Benign Disease Recurrence

While disease recurrence is most frequently considered in oncologic settings, liver transplant is, of course, most often employed in candidates with benign diseases. Pro-inflammatory states generally contribute to disease development indicating liver transplantation and then recurrence, as described for patients with alcoholic and non-alcoholic steatohepatitis (now termed metabolic dysfunction-associated liver disease (MASH)), and viral hepatitis B and C [83,84].

Specifically relating to biliary inflammation, we consider two common transplant indications as relevant: primary sclerosing cholangitis (PSC) and primary biliary cirrhosis (PBC). PSC is most common in Western countries at higher latitudes, with an incidence rate of 0.5–1.3 per 100,000 and a prevalence as high as 16.2 per 100,000 persons [85]. PBC occurs at similar rates, with estimated incidence and prevalence rates of 1.8 and 15 per 100 persons, respectively [86,87]. PSC is most commonly associated with inflammatory bowel disease (IBD), namely ulcerative colitis. As with PSC, PBC is most common in North America and Northern Europe, which report incidences roughly three times that of Southeast Asia [86,87]. Finally, benign infectious conditions such as hepatitis B (HBV) and hepatitis C (HCV) viruses are associated with biliary inflammation, though we focus on these least as their recurrence is well-established based on the use of post-transplant antivirals since the early 2000s [88,89,90]. The global seroprevalence of these two viruses is nearly 6% and is highest in the African region (HBV = 7.8% and HCV = 17.5%) [91].

PSC recurs in over one-quarter of patients after LT and has been clearly linked to pro-inflammatory states [92]. Along these lines, previous colectomy to remove a source of systemic inflammation is well-described as a strategy to prevent PSC recurrence [93]. However, there is also data to suggest that the inclusion of ileal pouch–anal anastomosis (IPAA/J-Pouch) is associated with a higher rate of PSC recurrence, which is theorized to be secondary to more inflammation in the pouch itself (frequently termed “pouchitis”) (Table 1) [94,95].

There is a uniquely interesting problem in PSC related to graft selection. Candidates with PSC are generally younger and medically healthier, and are often transplanted at lower laboratory MELD scores than the average liver recipient [93]. Such recipients are therefore more likely to receive DCD or extended criteria DBD grafts. Such organs come with a greater risk of inflammation due to higher degrees of IRI, and greater levels of mitochondrial injury and dysfunction [43,101,102]. This inflammatory state is seen clinically with a higher rate of acute cellular rejection (ACR) in DCD grafts [103,104], one contributor to biliary damage and inflammation. Frequent ACR occurrence is also the direct result of greater IRI, with innate immune signaling due to the already sustained inflammatory damage before the actual episode of rejection is identified. As discussed, the mitochondrial damage is prominent in the biliary system, which is also the specific source of hepatic pathology in these recipients. While these young, healthy liver recipients may tolerate the greater reperfusion injury in the short term, consideration might be given to minimizing the use of these grafts in PSC patients to prevent disease recurrence, particularly as these otherwise healthy and young patients might have the greatest life years gained if recurrence can be avoided. However, graft availability is an ongoing concern. Thus, alternatively, if grafts likely to experience greater degrees of IRI must be used, then strategies to prevent inflammation (such as hypothermic oxygenated perfusion—HOPE) should be strongly considered.

Another relevant liver disease is PBC, though generally less frequently studied in this context. Reported recurrence rates range from 9 to 35% [105]. Cholestasis is an identified risk factor for recurrent PBC, which is also reasonably well-linked to ongoing biliary inflammation and damage [106]. A clear link between ischemic damage and recurrent PBC has not yet been established, but it is logical that similar principles might apply in this population.

The biliary inflammation may finally play a role in other autoimmune liver diseases, including autoimmune hepatitis (AIH), viral cholangiopathy [107], and more, though we cannot conclusively claim this mechanistic link at the present time. Pro-inflammatory states and ACR both increase the risk of recurrent AIH, supporting a similar message that inflammation leads to worse immuno-proliferative hepatic environments that must be critically avoided in the peri-transplant period [108]. These recipients are also frequently kept on low-dose daily steroids to avoid rejection and the associated risk of disease recurrence secondary to chronic inflammation.

## 4. Clinical Strategies, Novel Interventions, and Future Perspectives

Case volumes of LT for CCA are significantly lower than those of HCC, and there has not been a well-established clinical link between ischemic/inflammatory complications and increased recurrence. Indeed, most of the research on the topic has, very appropriately, focused on patient selection and pre-transplant treatment to minimize recurrence. This has yielded important results, with the Mayo protocol identifying a proper patient population [109], and subsequent studies further refining optimal neoadjuvant chemo-radiation strategies [34,110,111].

However, there is clinical evidence that individual systemic inflammatory status impacts iCCA recurrence after R0 resection, which can conceivably be a harbinger for similar future findings in the transplant cohort (Figure 3) [112]. We herein discuss novel therapeutics, new approaches, and ongoing research into how to mitigate recurrent diseases and also improve post-transplant monitoring.

In transplantation, improvements can be made by treating the donor/graft, the recipient, and/or optimizing surgical techniques. In the study of HCC, our group and others have reported on mechanisms by which HOPE can mitigate mitochondrial inflammation, reduce the risk of NAS, and improve patient outcomes [32,38,43,101,113,114]. More specifically, the reintroduction of oxygen under cold conditions leads to a significant reduction in mitochondrial injury, limits the generation of ROS associated with succinate metabolism, and prevents the cytosolic release of the NDUFS-1 sub-unit of complex I [38]. This was even quantified using FMN, specifically noting a three- to eightfold reduction in mitochondrial damage with HOPE versus normothermic re-oxygenation [38].

Importantly, these mechanistic alterations with HOPE also lead to very clear clinical improvements in biliary outcomes [54,113,115,116]. HOPE has been shown to mitigate the risk of portal fibrosis and prevent the development of NAS in DCDs, and older and extended criteria DBD grafts, [117,118]. In contrast, NMP, has been associated with increased cell damage responses followed by biliary proliferation [119]. Though not directly linked to NMP modality or devices, these proliferative mechanisms have been correlated to more biliary strictures in non-perfused grafts, indicating where HOPE might be specifically beneficial [41].

Given the strength of both the mechanistic and clinical findings, it is reasonable to assume that treatments that mitigate biliary inflammation will also mitigate inflammation-mediated tumorigenesis and the recurrence of benign biliary disease after transplantation. This offers a very powerful and exciting potential in transplant oncology. As practitioners, this gives an entirely new area in which our intervention can reduce recurrence. Namely, we can improve the graft itself, rather than simply treating or selecting a recipient. At present, evidence suggests that these strategies should include the HOPE treatment of liver grafts where appropriate, as well as the use of FMN as a viability marker to assess riskier grafts with very likely high metabolic disruption. The former has a direct impact on graft quality, as described, with HOPE already theorized to be particularly helpful in CCA by Patrono and colleagues [120]. HOPE is estimated for FDA approval in late 2024/early 2025 in the US, and is currently available in most non-US transplant systems. HOPE is a practical and relatively inexpensive mechanism for graft preservation that seems to improve the graft as well. In short, HOPE mitigates inflammation and can reduce the incidence of disease recurrences that are linked to inflammation, including for CCA, PSC, and PBC.

**Figure 3 cancers-16-02789-f003:**
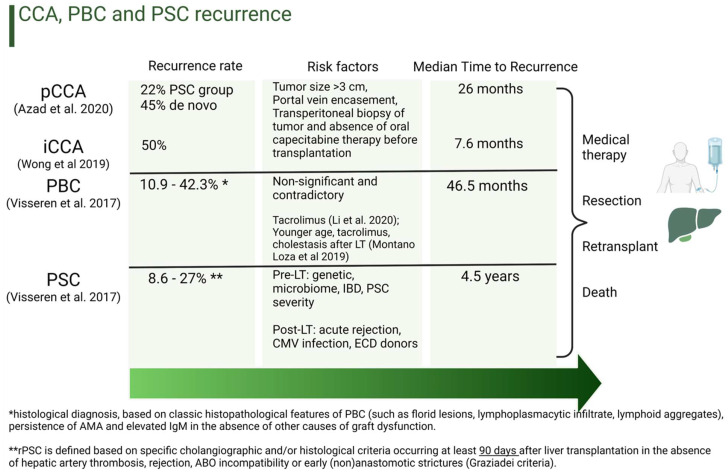
Risk Factors for Recurrence of Malignant and Benign Biliary Conditions. The median time for recurrence in CCA, PBC, and PSC is highly variable, as reported by different studies. Contributing risk factors, recurrence rates, and timings have also been described as contradictory and often non-significant. While PSC recurs at a median time of 4.5 years after liver transplantation, Graziadei et al. [121] highlight that an early recurrence within just 90 days could potentially be mistaken for progressively evolving non-anastomotic biliary strictures (NAS), which typically occur within the first year after liver transplantation. When recurrence occurs, the available options include medical therapy, such as chemo/immunotherapy for cancer, if the patient is suitable, resection in selected cases, and retransplantation for benign disease recurrence (i.e., PSC). pCCA: perihilar cholangiocarcinoma; iCCA: intrahepatic cholangiocarcinoma; PBC: primary biliary cirrhosis; PSC: primary sclerosing cholangitis; LT: liver transplantation; IBD: inflammatory bowel disease; CMV: cytomegalovirus; ECD: extended criteria donors; AMA: antimitochondrial antibody [33,34,35,106,122].

The latter, graft assessment with FMN, is suggested for all DCDs undergoing machine perfusion, aiming to avoid the transplantation of livers that are predisposed to develop cholangiopathy [43,47]. However, FMN might be even more critical as a marker in transplant for CCA. Specifically, we hypothesize that grafts with moderate levels of FMN, which indicate some predisposition, may be useful for transplantation, but perhaps could be used in a non-cancer case. This might avoid the introduction of biliary inflammation into grafts where such inflammation would bring a double hit: cholangiopathy and cancer recurrence.

We acknowledge that this recommendation is not yet based on robust data. It has, however, strong mechanistic support, but translational and clinical studies must be undertaken to confirm the relative impact of biliary inflammation, its reduction and assessment during machine perfusion, and its relationship with oncologic outcomes. This work, then, serves as a call for increased research and consideration of these factors in both the clinical and experimental spaces. Additionally, liver transplantation is increasingly exciting as a treatment for even newer indications, such as colorectal liver metastases (CRLM) and neuroendocrine tumors (NET), and translational research might investigate how the mitochondrial metabolism contributes to recurrence in these scenarios [123]. Improved graft preservation techniques might also reduce oncologic risk, allowing for the expansion of indications in liver transplant oncology [23,50].

In addition to the aforementioned methods of mitigated inflammation-associated recurrence risk, there are a few described pharmacologic interventions of potential utility. N-acteylcysteine has been associated with the attenuation of IRI in recent studies through the reduction of circulating cytokines [124,125]. NAC can be given in the donor, the recipient, or both, which demonstrates its versatility in mitigating IRI [125]. There is a described role for urso-deoxycholic acid (UDCA) prophylaxis in PBC to sequester toxic bile salts [126]. Antiviral medications are the most critical in preventing the recurrence of viral hepatitis and have been well shown as the most important factor in this effort [88,89,90]. The new direct-acting antivirals (DAAs) for HCV have led to a radical reduction in HCV-associated LT in modern practice and are also the key to preventing recurrence in the few remaining cases [127,128]. Finally, immunosuppression (IS) may be adjusted as well. Specifically, the addition of mTOR inhibitors have been well demonstrated to reduce the risk of cancer recurrence in both HCC and CCA [129]. Interestingly, it might seem that increasing IS may prevent the recurrence of PSC or PBC, given their autoimmune etiology, but this has not borne out in practice, and IS adjustments have not been shown to be beneficial [92,130].

There is also increasing research into novel monitoring tools for cancers after liver transplantation for cancer. This includes traditional biochemistry/tumor markers and imaging, in addition to more novel approaches such as circulating tumor DNA (ctDNA), the refinement/expansion of biomarkers, and more [131,132,133,134,135]. Most such work has focused on HCC, given its higher prevalence, but there are increasing reports of potential utility in CCA. While benign conditions are typically not monitored for recurrence in the same way, there are identifiable serologic changes that may be useful in monitoring for recurrence, which could be treated with the alteration of IS or improved biologic control, though, again, evidence on this topic is preliminary [92,136,137]. For example, cell-free DNA (cfDNA) has been shown to predict graft injury in the context of rejection, but this same mechanism might apply to injury from inflammation in the more acute setting of IRI [138]. Like ctDNA for oncologic cases, cfDNA might have utility in the future in the monitoring of benign inflammation-associated cases such as PSC or PBC, and its donor-specific nature might help distinguish between liver inflammation and other systemic manifestation such as in the GI tract.

## 5. Conclusions

Ischemia-reperfusion injury and related inflammation after liver transplantation are the downstream manifestations of mitochondrial damage. This injury leads to ischemic cholangiopathy and has been separately linked to an increased risk of cancer recurrence, as the mitochondrial damage is most concentrated in the biliary system. Thus, we propose that strategies to mitigate or prevent biliary inflammation, such as graft assessment with FMN or treatment with hypothermic oxygenated perfusion (HOPE) might be most critical in transplants for cholangiocarcinoma. Future studies should focus on this in both a translational and clinical sense, as this might shed light on methods to prevent the recurrence of CCA, and even help improve outcomes for liver transplants for other cancers, such as colorectal liver metastases and neuroendocrine tumors.

## Figures and Tables

**Figure 1 cancers-16-02789-f001:**
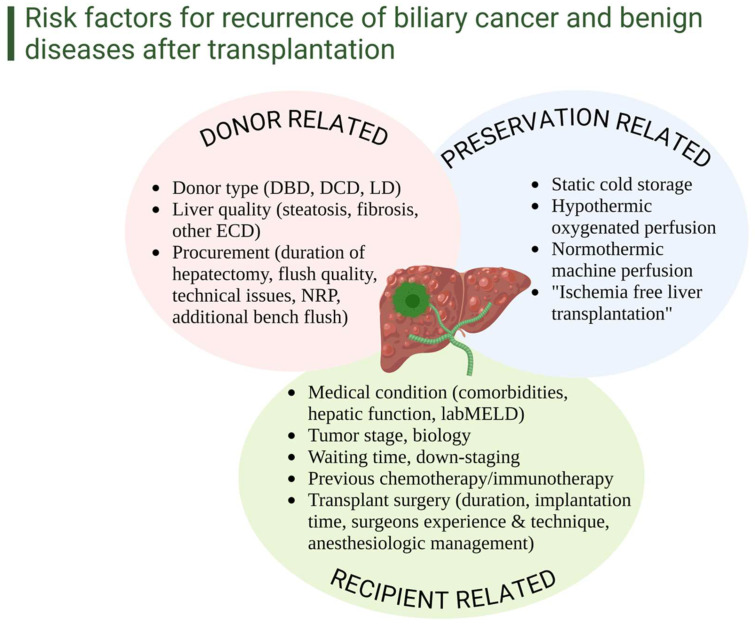
Clinical risk factors for recurrence of biliary cancer and benign biliary conditions after liver transplantation. Many risk factors may contribute to the recurrence of biliary cancer and benign biliary diseases after transplantation, including donor-related factors such as higher-risk grafts, like DCD, and procurement factors, like the duration of hepatectomy and flush quality. Additionally, the type of preservation could significantly impact disease recurrence, potentially mitigating ischemia-reperfusion injury. In this context, HOPE has shown a protective effect on mitochondrial injury, with significantly lower IRI-associated inflammation compared to normothermic techniques. Early clinical studies demonstrate that endischemic HOPE reduces the HCC recurrence rate with better recipient survival. Randomized controlled trials are starting in Europe, comparing HOPE with static cold storage in liver transplants for HCC candidates. Lastly, recipient-related factors contribute significantly to recurrence, including tumor stage and biology, overall medical recipient status, history of chemotherapy and immunotherapy, and the technical aspects of the transplant surgery. DBD: donation after brain death, DCD: Donation after Circulatory Death, ECD: extended criteria donor, LD: living donor, NRP: normothermic regional perfusion, MELD: model for end-stage disease.

**Figure 2 cancers-16-02789-f002:**
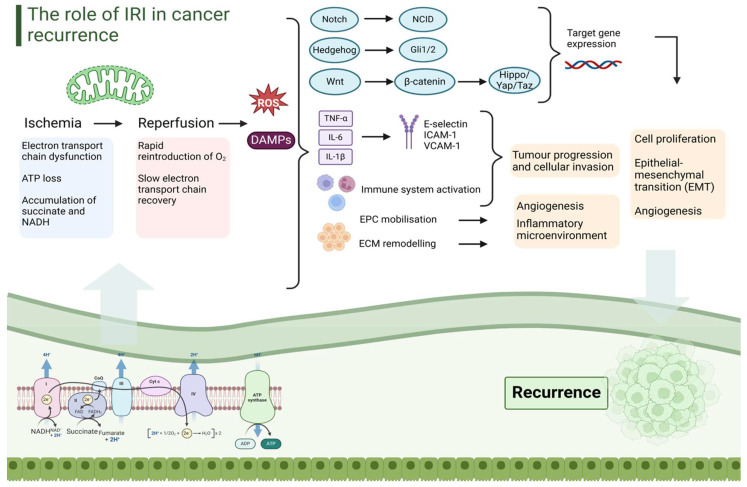
The Role of Ischemia-Reperfusion Injury in Cancer Recurrence. Ischemia-reperfusion injury (IRI) plays a fundamental role in cancer recurrence. During reperfusion, reactive oxygen species (ROS) and damage-associated molecular patterns (DAMPs) are released in large quantities, initiating a downstream cascade. This cascade activates the Notch, Hedgehog, Wnt/β-catenin, and Hippo/Yap/Taz pathways, which are critical for cell proliferation, epithelial–mesenchymal transition (EMT), and angiogenesis. Additionally, there is significant cytokine release including TNF-α, IL-6, and IL-1β. Such molecules, by activating E-selectin, ICAM-1, and VCAM-1, contribute to the establishment of an inflammatory microenvironment, supporting tumor progression, resettling, and invasion of cTCs. Concurrently, further downstream, reperfusion injury activates the innate immune system, which actively participates in these pathways. Ultimately, this promotes endothelial progenitor cell (EPC) mobilization and extracellular matrix (ECM) remodeling, leading to angiogenesis and further promoting the ongoing inflammation.

**Table 1 cancers-16-02789-t001:** Donor or preservation-related risk factors for recurrence of biliary diseases after liver transplantation.

Authors and References	Disease	Country; Region; Center	Number of Patients	Donor Type	Recurrence Rate	Donor or Preservation-Related Risk Factors
Catanzaro et al. 2024 [96]	PSC	Padua, Italy	33	DBD	27%	CIT Female donor
Steenstraten et al. 2019 [97]	PSC	Japan, USA, UK, Hungary, Germany, Canada, Nordic countries	2159	DBD, LD	17.7%	Donor age
Alabraba et al. 2009 [98]	PSC	Birmingham, UK	230	DBD	23%	ECD grafts
Gordon et al. 2016 [99]	PSC	North America (US and Canada)	307	DBD, DCD, LD	11%	Donor age
Abu-Elmagd et al. 1997 [100]	PBC	-	421	DBD	11%	CIT

No studies reported on specific donor or preservation-related risk factors for recurrence of cholangiocarcinoma. Parameter cut-offs for higher risk are entirely lacking. CIT: cold ischemia time; DBD: donation after brain death; DCD: Donation after Circulatory Death; ECD: extended criteria donor (includes DCD organs, macrosteatosis, and prolonged cold ischemia times, for example); PBC: primary biliary cirrhosis; PSC: primary sclerosing cholangitis.

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
