# Peer review of "The Impact of Biliary Injury on the Recurrence of Biliary Cancer and Benign Disease after Liver Transplantation: Risk Factors and Mechanisms"

_cancers, 2024, doi:10.3390/cancers16162789_

Round 1
Reviewer 1 Report
Comments and Suggestions for Authors
The main question addressed in this review article is to assess and describe the underlying mechanisms associated with biliary inflammation and biliary pathology after liver transplantation. The review is of practical significance. Post transplant many of these patients may develop recurrent disease.
The authors propose that strategies to mitigate or prevent biliary inflammation in particular for cholangiocarcinoma cases. It provides an excellent comprehensive discussion from a world-renowned transplant center, that will be of interest to all transplant surgeons along with a spectrum of other disciplines; oncologists, hepatologists, governmental funding agencies and pharmaceutical scientists.
It should be considered doing a co-operative study focusing on IRI associated inflammation with other interested centers.
To prevent recurrence of CCA following liver transplantation as well as outcomes for other cancers you encourage centers to simulate your experience. Have you interacted with other centers who have the capacity to do so but have not done so?
Regardless your center's experience is commendable'
The references are appropriate, properly displayed and timely.
The Figures are well done and helpful.
Author Response
Please See attached word document.

Reviewer 2 Report
Comments and Suggestions for Authors
This is an outstanding review which addresses the underlying mechanisms associated with biliary inflammation and biliary pathology after liver transplantation. the topic is intriguing and the review is very well written and presented. Figures are also very good.
overall, I feel that this manuscript will be a major addition to the field. just one comment, in the sake of completness. I understand this is a narrative review, but can the Authors comment on how papers and evidence were searched for and selected for inclusion in this manuscript?
Comments on the Quality of English Languagevery good, only some proof-reading is necessary.
Author Response
please see attached word document

Reviewer 3 Report
Comments and Suggestions for Authors
This work is well conducted and informative, while certain modifications should be made to make it acceptable.
1. The authors should describe the different incidences of various diseases and etiologies in various regions.
2. The author should describe infectious diseases more thoroughly and the associated mechanisms that induce these diseases.
3. The authors should put more discussion on how to reduce the recurrent diseases and potential/novel therapeutics and monitoring tools.
Author Response
please see attached word document.

Round 2
Reviewer 3 Report
Comments and Suggestions for Authors
The revision is accepted for publication.